# Minimally Invasive versus Open Distal Pancreatectomy in the 2020s: Recent Institutional Experience and a Narrative Review of Current Evidence

**DOI:** 10.3390/jcm12206578

**Published:** 2023-10-17

**Authors:** Saad Rehman, Ishaan Patel, David Bartlett, Darius Mirza

**Affiliations:** 1Liver Transplant and HPB Unit, Queen Elizabeth University Hospital Birmingham, Birmingham B15 2GW, UK; 2Institute of Immunology and Immunotherapy, University of Birmingham, Birmingham B15 2TT, UK; 3Hon Professor of HPB and Transplant Surgery, University of Birmingham, Birmingham B15 2TT, UK

**Keywords:** pancreatic surgery, distal pancreatectomy, surgical outcomes, pancreatic cancer

## Abstract

(1) Background: Distal pancreatectomy is a standard treatment for tumours of the pancreatic body and tail. Minimally invasive techniques for all types of pancreatic tumours (benign and malignant) are being established, while concerns regarding oncological safety, cost effectiveness and learning curves are being explored with prospective studies. This paper presents our unit’s data in the context of the above concerns and provides a relevant narrative review of the current literature. (2) Methods: Data were collected retrospectively between 2014 and 2021 for all adult patients who underwent elective distal pancreatectomy in our tertiary care referral HPB Unit. Data on demographics, underlying pathology, perioperative variables and post-operative complications were collected and reported using descriptive statistics. On review of the Miami guidelines, four important but less validated areas regarding distal pancreatectomy are presented in light of the current evidence; these are recent randomised controlled trials, oncological safety, cost effectiveness and learning curves in minimally invasive distal pancreatectomy (MIDP). (3) Results: 207 patients underwent distal pancreatectomy in total from 2014–2021, with 114 and 93 patients undergoing open and minimally invasive techniques, respectively. 44 patients were operated on for PDAC in the open vs. 17 in the minimally invasive group. The operative time was 212 min for the open and 248 min for the minimally invasive group. The incidence of pancreatic fistula was higher in the minimally invasive group vs. the open group (16% vs. 4%). (4) Conclusions: Our unit’s data conform with the published literature, including three randomised control trials. These published studies will not only pave the way for establishing minimally invasive techniques for suitable patients, but also define their limitations and indications. Future studies will inform us about the oncological safety, cost effectiveness, overall survival and learning curves regarding patients undergoing minimally invasive distal pancreatectomy.

## 1. Introduction

Minimally invasive surgery has replaced open surgery as the gold standard in various surgical specialties. The first minimally invasive pancreatic surgery was reported in 1996 by Gagner [1]; however, the uptake of minimally invasive techniques has been relatively slow due to factors such as increased cost, technical expertise and the naturally challenging anatomy of the peripancreatic region. Distal pancreatectomy (DP) is indicated for tumours of the body and tail of the pancreas. Pooled data from several case series and propensity-matched comparison studies state the efficacy and non-inferiority of minimally invasive distal pancreatectomy (MIDP) [2]. In the last 5 years, owing to great initiatives in research, collaboration and international consensus, there has been commendable advancement in our understanding of the feasibility of laparoscopic distal/left sided pancreatectomy. MIDP has been an area of active research interest, and synthesised live and most up-to-date research evidence can be accessed through websites such as evidencemap.surgery [3]. High-quality randomised controlled trials have shown that MIDP is better than open distal pancreatectomy (ODP) in terms of time to functional recovery, delayed gastric emptying (DGE), blood loss and duration of hospital stay [4,5]. This emerging body of evidence also matches the widening and gradual acceptance of laparoscopic distal pancreatectomy in the surgical community. The Miami International Evidence-based Guidelines on Minimally Invasive Pancreas Resection established that ‘MIDP for benign and low-grade malignant tumours is to be considered over ODP since it is associated with a shorter hospital stay, reduced blood loss, and equivalent complication rates’ [6]. There remained concerns as to whether MIDP would furnish comparable oncological outcomes in distal pancreatic resections performed for cancer, and it was thought that awaited randomised trials from Europe and Asia would be able to fill this evidence gap in the near future. The cost of minimally invasive alternatives has to be considered when deciding whether these less invasive techniques justify the gains they provide for patients, especially in finance-restricted parts of the world. Minimally invasive pancreatic surgery has gained momentum in high-volume centres, but the learning curve for surgeons requires more work to inform a wider surgical group of the expected technical challenges and associated early outcomes in smaller centres around the world.

This review aims to present our unit’s experience of distal pancreatectomy cases and presents the current state of the evidence in four important areas: recent randomised controlled trials, oncological safety, cost effectiveness and learning curves in minimally invasive distal pancreatectomy.

## 2. Materials and Methods

All adult patients who underwent elective distal pancreatectomy from 2014–2021 for benign or malignant tumours were included. Data were retrieved with the help of a proforma that included variables for the study including patient demographics (age, gender, BMI, comorbidities) and perioperative details (duration of operation, hospital stay). Descriptive statistics were used to represent simple data: qualitative variables as percentages or proportions and quantitative variables as medians and range. Statistical analyses were undertaken using Statistical Package for the Social Sciences (SPSS) version 24.0 (SPSS Inc., Chicago, IL, USA).

The systematic search of the literature was performed for the period between 2010 and 2023. The search was designed to identify all randomised controlled trials which compared MIDP vs. ODP, as well as relevant systematic reviews, which were then thoroughly discussed and scrutinised. The PubMed, Medline, Scopus, Embase and Cochrane Library databases were screened. The search included both free text and the MeSH terms: ‘Pancreatic neoplasm’, ‘Pancreatic Cancer’, ‘Pancreatic ductal adenocarcinoma’, ‘Pancreatic adenocarcinoma’, ‘Left pancreatectomy’, ‘Distal Pancreatectomy’, ‘Pancreatectomy’, ‘Laparoscopy’, ‘Laparoscopic’, ‘Minimally invasive’, ‘cost effectiveness’ and ‘learning curve’. Once the search was concluded, the references for the retrieved articles were checked manually for further studies and any disagreement between the authors was resolved by consensus. A narrative and clinically relevant description was given and, where appropriate, summarised in tables.

## 3. Results

### 3.1. Retrospective Data from Our Unit

During the study period (2014–2021), a total of 211 patients underwent distal pancreatectomy. Four patients were excluded as the indication was non-elective (trauma). The numbers of patients undergoing open and MIDP were 114 (55%) and 93 (45%), respectively. Forty-four (39%) patients were operated for pancreatic ductal adenocarcinoma (PDAC) in the open vs. 17 (18%) in the minimally invasive group (*p* = 0.001). Eleven patients in each group had spleen-preserving surgery. The operative time was 212 min (range, 91–519 min) for open and 248 min (range, 139–427 min) for MIDP. Nineteen (17%) patients in the open group had another organ (stomach, colon, small bowel, etc., and excluding spleen and adrenals) included in the resection, compared to only one in the minimally invasive group. None of the patients in the minimally invasive group underwent vascular resection, whereas five patients (4%) in the open group had vascular resection. Although overall complications were the same in both groups, grade B/C post-operative pancreatic fistula (POPF) was higher in the minimally invasive group (16% vs. 4%, *p*: 0.003). The post-operative hospital stay was considerably shorter in the minimally invasive group (5 days) vs. open (8 days). The baseline, operative and post-operative characteristics of these patients are illustrated in Table 1.

### 3.2. Narrative Review of Randomised Controlled Trials

There have been several cohort studies but only three published randomised controlled trials comparing open and minimally invasive distal pancreatectomy [4,7]. All three trials (LEOPARD 2019, LAPOP 2020, DIPLOMA 2023) were good-quality studies and from European centres; the LAPOP trial (unicentre, unblinded) compared open with laparoscopic DP and the LEAOPARD trial (multicentre, patient blinded) compared open with both laparoscopic and robotic DP in all cases of pancreatic tumours (benign/low malignant and malignant lesions). Both trials demonstrated that patients who had MIDP stayed in hospital two days fewer than those who underwent ODP. The LEOPARD trial showed a quicker time to functional recovery following MIDP compared to ODP and the rest of the perioperative variables were comparable. The long-term follow-up data for both these studies along with quality-of-life outcomes have also recently been published. The LAPOP trial long-term data confirmed that there was a significant improvement in quality of life in the MIDP group and that the difference persisted after 2 years post-surgery [8]. The third randomised controlled trial (DIPLOMA) comparing oncological outcomes specifically in pancreatic ductal adenocarcinoma (PDAC) has been recently published [8]. This is an international, multicentre, patient- and pathologist-blind randomised controlled trial with the primary end point of radical resection (R0, ≥1 mm free margin). An R0 resection occurred in 83 (73%) patients in the MIDP group and in 76 (69%) patients in the ODP group (pnon-inferiority = 0.039). This shows that the MIDP group has a higher percentage of patients in which complete and tumour-free resection was achieved. The median lymph node yield was comparable (22.0 [16.0–30.0] vs. 23.0 [14.0–32.0] nodes, *p* = 0.86). Intraperitoneal recurrence was also similar (41% vs. 38%, *p* = 0.45), as was survival in both the open and the MIDP groups. Data from these three randomised controlled trials (RCTs) are summarised in Table 2, Table 3, Table 4 and Table 5, including basic characteristics, treatment protocols, quality variables and results data. There are at least five more RCTs currently in the recruitment phase and these are listed in Table 6.

### 3.3. Minimally Invasive Distal Pancreatectomy (MIDP) for Pancreatic Ductal Adenocarcinoma

The Miami guidelines in 2020 stated the feasibility of MIDP in cases of PDAC but recommended prospective randomised studies to further build the evidence base. This evidence of feasibility was generated mainly from non-randomised studies. Six systematic reviews based on cohort series and propensity-score-matched analyses have been published comparing open and MIDP in PDAC [9,10,11,12,13]. Outcomes measured included pathological (R0 resection, lymph node yield), surgical (operation time, intraoperative bleeding, post-operative complications, hospital stay), recurrence rates and overall short- and long-term survival. Initial systematic reviews showed that overall survival was comparable in the open and MIDP groups but that there was still uncertainty regarding the oncological efficacy. Yang et al. were able to confirm that the risk of a *positive* resection margin was lower in the MIDP group and that recurrence rates were similar [11]. Further data also confirmed similar results in terms of overall survival, perioperative and surgical outcomes, but Tang et al. showed that fewer nodes were retrieved in the MIDP group [14]. There was also a concern that patients in the MIDP group had less perineural and lymphovascular involvement. This very rightly represents the apprehension and possibly bias in case selection for the MIDP group to include smaller and less invasive tumours. These concerns regarding adequate lymph node yield and resection margins were addressed to a good extent by the DIPLOMA trial (published August 2023), which is the only randomised controlled trial to date in this area. It has shown better R0 resection rates in the MIDP group and comparable results to the ODP group in terms of lymph node yield and recurrence rates. Data from ongoing RCTs will help establish the evidence base for short- and long-term outcomes for PDAC. A list of ongoing trials investigating this is provided in Table 6 [15].

### 3.4. Cost Effectiveness of MIDP

Financial implications are usually the first hurdle to cross for a developing surgical innovation once clinical safety has been established. MIDP has been proven to reduce hospital stay by 2 days and this should compensate for the higher associated costs. A recent systematic review reported significant costs for complications associated with distal pancreatectomy and suggested that hospital stay was the main contributor to cost [16]. Cost evaluation from the LAPOP randomised controlled trial after a 1-year follow up showed laparoscopic distal pancreatectomy (LDP) to be the most cost-effective [17]. Another systematic review including 16 studies and about 2400 patients showed similar results to those of the above RCT and demonstrated that LDP and robotic DP were both better than ODP but LDP was most likely to be cost-effective and safe [18]. Further prospective studies will be required to investigate cost with relation to quality of life over a longer follow-up period to establish the absolute cost effectiveness of MIDP.

### 3.5. Learning Curves in MIDP

Cost effectiveness, surgical safety and the patient’s overall outcomes are associated with the surgeon’s experience and volume of work [19]. Improved outcomes from MIDP come at a cost of technical challenges and a relatively long learning curve. The literature on learning curves in MIDP was initially single-centre and single-surgeon/group studies [20,21]. The learning curve for MIDP had initially been reported to be as little as 10–17 procedures. These studies have gradually become less relevant in the current era for several reasons: the reported studies typically described initial experiences of MIDP from high-volume centres with considerable bias; there has been increased acceptance of MIDP in the wider surgical community, including smaller centres; and there are evolving indications for MIDP that now also include malignant tumours. In a nationwide UK study, the learning curve was estimated to be 30 cases [22]. A large retrospective cohort study including over 2600 patients from multiple high-volume European centres presented data regarding the cut-off competency point for different variables in distal pancreatectomy [23]. The study described different estimated break points for different variables: the break point estimated for conversion was 40 procedures (95% CI, 11–68 procedures); for operative time, 56 procedures (95% CI, 35–77 procedures); and for intraoperative blood loss, 71 procedures (95% CI, 28–114 procedures). The textbook outcome (the absence of grade B/C POPF, PPH, bile leakage, major complications, readmission and in-hospital mortality) was, however, achieved only after 85 procedures had been completed. This study was a great collaborative initiative and considered more variables (including textbook outcome) against the learning curve. We would, however, point out that most studies, including the latter, do not address the direct relationship between learning curve and procedure difficulty (larger tumour size, neoadjuvant chemotherapy, etc.). Intraoperative blood loss and operative time have frequently been reported as surrogates, but their relationship with procedure difficulty has not been fully established. This is becoming more and more relevant as the indications for MIDP are being pushed and are now more inclusive of a wider and more complex pathology. Further studies should emphasise the uniformity of the terminology used to describe complications, to help investigate cost and training implications better and to formulate assessment methodologies to train future surgeons.

## 4. Discussion

Minimally invasive distal pancreatectomy (MIDP) is an established strategy for benign and premalignant or low-malignant potential tumours of the body and tail of the pancreas. Evidence is also accumulating for its feasibility and comparability for malignant lesions, such that increasing numbers of patients with malignant tumours are being offered MIDP.

Overall, MIDP has shown equivalent outcomes compared to the open approach, with the added advantage of shorter hospital stays and the potential for a quicker return to functional recovery. There may be an associated increase in low-grade POPF, which is likely to continue to decrease as familiarity with and wider adoption of MIDP progresses. Our unit’s data from the last 8 years conform with published larger prospective data in terms of shorter duration of hospital stay and comparable perioperative outcomes for MIDP, even though there may be differences in the patient and disease populations in the two groups—for example, a much smaller proportion of patients with PDAC undergoing MIDP, and vascular resections being performed only in patients undergoing open surgery. The lower proportion of PDAC patients (only 14.9%) in the minimally invasive group in our unit attests to the earlier concerns regarding oncological safety in distal pancreatectomy. As regards the somewhat higher rates of low-risk POPF seen in our unit’s experience, this may be related to the different case mix in the open vs. MI groups, or might reflect surgical learning curves for MIDP. Moreover, it matches an earlier nationwide UK study that reported similar POPF outcomes in the MIDP group and was thought to be perhaps due to a higher prevalence of mucinous cystic neoplasms or neuroendocrine tumours (which are more likely to cause POPF) in the MIDP group.

We identified four areas in which either the literature was lacking or a significant advancement had been made since the Miami guidelines (2020) were published, and presented the most up-to-date narrative review. A summative and tabulated review of three published randomised controlled trials has been presented, as well as a list of ongoing RCTs awaiting publication. Oncological safety has been discussed in the context of the DIPLOMA trial. Cost effectiveness is increasingly relevant with widening acceptance of robotic techniques in MIDP. There is a developing consensus that costs related to pancreatic surgery are dependent on the duration of hospital stay and post-op complications. This forms the basis of the argument for increasing the use of minimally invasive techniques. Learning curves associated with MIDP have been formulated using varying criteria. The most recent article, published by a Dutch group, reports that, although an individual outcome-based learning curve may be achieved following a relatively small number of procedures, if textbook outcome is the considered criteria, the number of procedures to achieve this is much higher, i.e., 85 procedures [23]. Further studies should report not only the textbook outcomes but also provide a comparison with the open technique, which would be more relevant.

We acknowledge and appreciate the three randomised trials published so far in distal pancreatic resection. The data on oncological feasibility initially were inconclusive regarding lymph node yield and negative resection margins, although survival and other surgical outcomes were comparable. More recent systematic reviews have shown somewhat comparable data in both OPD and MIDP groups overall, but uncertainty remains as to how comparable and sometimes better surgical outcomes can be combined with oncological safety. The trend in systematic reviews over the last decade shows the progression of data gradually turning in favour of MIDP. The recent DIPLOMA trial published in the Lancet is the first randomised study to assess oncological outcomes of MIDP in cancers. It showed non-inferiority in the rate of R0 resection and lymph node yield and comparable perioperative outcomes. We do note that the mean tumour size in the DIPLOMA trial was just 30 mm (with the largest being 42 mm). Questions remain about the applicability of results to centres with smaller caseloads and less experienced surgeons, as well as the role of MIDP for larger tumours, those requiring vascular resection and reconstruction and surgical resection following downstaging neo-adjuvant treatment.

The strength of this article is that it highlights important research areas in minimally invasive distal pancreatectomy based on the existing literature. The article is limited by the current lack of good-quality prospective studies in this field.

## 5. Conclusions

In conclusion, this review highlights the current evidence for MIDP in benign, low-malignant tumours and small resectable pancreatic cancers. Future prospective trials should focus on the further optimisation of outcomes following MIDP and its use with more advanced pancreatic tumours.

## Figures and Tables

**Table 1 jcm-12-06578-t001:** Baseline and perioperative characteristics of included patients.

	ODP	MIDP
Number of Patients	114	93
Age (Mean in years)	62	58
Female Gender n (%)	47 (41)	52 (56)
Spleen preserving	11 (10)	11 (12)
PDAC/mets	44 (39)	17 (18)
Benign/low malignant Tumours	65 (57)	74 (80)
Chronic Pancreatitis	5 (4)	2 (2)
Conversion to open		15 (16)
RAMPS (radical antegrade modular pancreatosplenectomy)	4 (4)	13 (14)
Appleby Procedure	3 (3)	0
Operative Time (median in min)	212	248
Patients requiring intra-op blood transfusion	15 (13)	0
Adrenal Gland resection	15 (13)	8 (9)
Additional Organ resection (colon, kidney, stomach, small bowel)	19 (17)	1 (1)
Vascular resection	5 (4)	0
All complications	41 (36)	44 (47)
Grade I/II CD complications	35 (31)	33 (35)
Grade III/IV CD complications	8 (7)	7 (8)
Complication—no intervention needed	15 (13)	24 (26)
Complication—medical treatment	22 (19)	14 (15)
Complication—surgical/radiological intervention	4 (4)	6 (6)
POPF (postoperative pancreatic fistula) (Grade B/C)	5 (4)	15 (16)
30-day mortality	5 (4)	0
1-year mortality	15 (13)	3 (3)
Hospital stay, median (days)	8	5

**Table 2 jcm-12-06578-t002:** Characteristics of the randomised controlled studies.

Trial	Country	Nature of the Trial	Ethics Approval	Patients Number (ODP:MIDP)	Surgical Intervention	Primary Outcome
Abu Hilal 2023 (DIPLOMA Trial) [8]	Italy/Sweden	RCT (DIPLOMA Trial)	Yes	258 (127:131)	Open Distal Pancreatectomy: Minimally Invasive Distal Pancreatectomy	Microscopically free radical resection margin R_0_ (R_0_ ≥ 1 mm tumour-free resection margin)
Björnsson 2020 (LAPOP Trial) [5]	Sweden	RCT (LAPOP Trial)	Yes	58 (29:29)	Open Distal Pancreatectomy: Laparoscopic Distal Pancreatectomy	Length of postoperative hospital stay
De Rooij 2019 (LEOPARD Trial) [4]	Netherland	RCT (LEOPARD Trial)	Yes	108 (57:51)	Open Distal Pancreatectomy: Minimally Invasive Distal Pancreatectomy	Time to functional recovery post-surgery (days)

**Table 3 jcm-12-06578-t003:** Treatment protocol adopted in included trials.

Trial	ODP	MIDP
Abu Hilal 2023 (DIPLOMA Trial) [8]	Open Distal Pancreatectomy	Laparoscopic + Robotic Distal Pancreatectomy
Björnsson 2020 (LAPOP Trial) [5]	Open Distal Pancreatectomy	Laparoscopic Distal Pancreatectomy
De Rooij 2019 (LEOPARD Trial) [4]	Open Distal Pancreatectomy	Laparoscopic (42) + Robotic (5) Distal Pancreatectomy

**Table 4 jcm-12-06578-t004:** Quality variables of the included trials.

Trial	Randomisation Technique	Blinding	Allocation Concealment	Intention to Treat Analysis	Power Calculations	Trial Registration Number
Abu Hilal 2023 (DIPLOMA Trial) [8]	Central computerised simple sequence randomisation	Patient, pathologist blinded and Outcome assessor blinded	Sequentially numbered, opaque, sealed envelopes	Yes	Yes	ISRCTN44897265
Björnsson 2020 (LAPOP Trial) [5]	Central computer-generated block randomisation	Patient blinded and Outcome assessor blinded	Sequentially numbered, opaque, sealed envelopes	Yes	No	ISRCTN26912858
De Rooij 2019 (LEOPARD Trial) [4]	Permuted computer-generated block randomisation list	Patient blinded and Outcome assessor blinded	Non-transparent sealed envelope	Yes	Yes	NTR5689

**Table 5 jcm-12-06578-t005:** Perioperative variable data from the included trials.

Trial		Patients	No of Patients with PDAC n (%)	Op Time min (Range)	Blood Loss mL (SD)	Post-Op Stay Days (SD)	Complications (>Grade III CD)	Recurrence n (%)	Lymph Node Yield Median (Range)	R_0_ Resection n (%)	POPF	DGE	FU
Abu Hilal 2023 (DIPLOMA Trial) [8]	ODP MIDP	114 117	114 (100) 117 (100)	209.0 (158.0–257.0) 240.0 (175.3–308.8)	200.0 (100.0–400.0) 200.0 (100.0–300.0)	7.0 (6.3–7.7) 7.0 (6.4–7.6)	26 (20.5) 25 (19.1)	43 (38%) 48 (41%) *p* = 0.45	23 (14–32) 22 (16–30) *p* = 0.86	76 (69%) 83 (73%)	20 (17.5) 25 (21.4)	3(2.7) 2(1.7)	36 m
Björnsson 2020 (LAPOP Trial) [5]	ODP LDP	29 29	2 (6.8) 6 (20)	120 (11.5) 120 (8.75)	100 (50) 50 (31.25)	8 (1) 6 (0.75)	8/29 (27.5%) 4/29 (13.7%)	NA	NA	0/0 4/6	11/29 9/29	5/29 1/29	24 m
De Rooij 2019 (LEOPARD Trial) [4]	ODP MIDP	57 51	10 (18) 13 (25)	179 (25.5) 217 (35.5)	400 (143) 150 (75)	8 (1.5) 6 (0.75)	21/57 (36.8%) 39/51 (76.4%)	NA	14.25 11.5	4/10 7/13	13/57 20/51	1/57 3/51	44 m

**Table 6 jcm-12-06578-t006:** Ongoing RCTs in distal pancreatectomy for PDAC.

Identifier	Title	Expected Date of Trial End
NCT03957135	Laparoscopic versus open distal pancreatectomy for pancreatic cancer: a multicentre randomized controlled trial	30 November 2025
ISRCTN44897265	Distal pancreatectomy, minimally invasive or open, for malignancy	1 May 2024
KCT0004176	Multicentre prospective randomized controlled clinical trial for comparison between laparoscopic and open distal pancreatectomy for ductal adenocarcinoma of the pancreatic body and tail	30 November 2023
NCT03792932	Laparoscopic versus open pancreatectomy for body and tail pancreatic cancer	31 January 2022
ChiCTR1900024648	A randomized controlled study for the short-term oncologic outcomes of robot-assisted radical and open anterograde modular pancreaticosplenectomy	30 November 2020
DRKS00014011	Distal pancreatectomy of a randomized controlled trial to compare open versus laparoscopic resection (DISPACT 2-TRIAL)	Not reported
ChiCTR2000038933	Robotic versus open radical antegrade modular pancreatosplenectomy for pancreatic cancer of the body and tail: a multicentre, randomized controlled trial	Not reported

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
