# Peer review of "Minimally Invasive versus Open Distal Pancreatectomy in the 2020s: Recent Institutional Experience and a Narrative Review of Current Evidence"

_jcm, 2023, doi:10.3390/jcm12206578_

Round 1
Reviewer 1 Report
​Minimally invasive surgery has replaced open surgery as the gold standard in various surgical specialities. The first minimally invasive pancreatic surgery was reported in 1996 by Gagner however, the uptake of minimally invasive techniques has been relatively slow; due to factors such as increased cost, technical expertise and naturally challenging anatomy of the peripancreatic region. Distal pancreatectomy (DP) is indicated for tumours of the body and tail of the pancreas.
The title and content of the article represent a topic of real interest worldwide.
The subject of the study is topical with real interest for the future because there is great heterogeneity in the specialized literature regarding the indications of the minimally invasive approach in the case of distal pancreatic tumors. From what I have analyzed, there are no absolute gold standard criteria for the local, regional or biological order of distal pancreatic tumors in order to perform minimally invasive interventions. I appreciate your effort to expose this important component in terms of the indications of the minimally invasive approach, the economic benefits, the biological benefits, but nevertheless the analysis of the oncological criteria shows a poor exposure in the present study.
Maybe the extension of this study on a larger population sample could improve the statistical results and create certain gold standard criteria for inclusion in the minimally invasive surgical approach titrated according to tumor stage, tumor biology and the experience of each surgical center.
The introduction of the article presents originality by proposing a topic with a huge academic potential.
The bibliographic data inserted along the article presents a qualitative chronology. The subject of the article represents a true scientific revolution in its field.
The material and methods section of the article presents a quantitative and qualitative exposition of the research plan, respectively a good reproducibility in order to develop other studies with this theme.
The results of the article present a logical and chronological exposition outlining qualitative aspects of the benefit. The figures and tables keep a specific chronology throughout their exposition, presenting qualitative aspects related to the subject of the article.
The topic of the article is a real interest for the future with major importance in this field. I consider it necessary to develop new studies on this subject and implement them on a population scale. The article presents an important research point with an optimal linguistic exposition, having an exponential potential for the future. This present article is written in a clear and concise manner.
The article presents originality, with an optimal literary exposition, representing a topic of real interest for the future with objective results at the research level. The article represents a launching platform in its field and from the point of view of the characteristics it is included for publication.
Author Response
Dear Sir/Madam,
Please see the attachment with all the comments addressed point by point.
Kind regards,

Reviewer 2 Report
Overall, I like the submitted manuscript. It is not too difficult to read and provides useful information for physicians who encounter pancreatic lesions in their clinical practice.
I believe the article is intended for a wider readership and not just for abdominal surgeons, so I think it would be good to add an explanation of the facts from lines 120-122. The authors stated here that current surgical techniques will achieve radical resection (into healthy tissue) in 73% and 69% of patients, respectively. It is of great interest to the non-surgeon physicians (spec. to me :-)) to know how these patients are managed going forward, whether re-operation or other approaches are the solution. I think that a brief reflection on the possibilities of achieving radical resection in all operated patients and whether such an approach would not imply an unacceptable increase in the risk of complications would be useful in the discussion.
I know about the passion of English native speakers for abbreviations, but the clarity of the text for practitioners of other disciplines requires that all abbreviations used are explained in the text. Specifically, I mean:
PDAC, RAMPS, POPF, RTC
no comments - just explain shorts
Author Response

(The authors gave the same response as above.)
